# The dynamic interplay of host and viral enzymes in type III CRISPR-mediated cyclic nucleotide signalling

Januka S Athukoralage, Shirley Graham, Christophe Rouillon[†], Sabine Grüschow, Clarissa M Czekster, Malcolm F White*

Biomedical Sciences Research Complex, School of Biology, University of St Andrews, St Andrews, United Kingdom

**Abstract** Cyclic nucleotide second messengers are increasingly implicated in prokaryotic anti-viral defence systems. Type III CRISPR systems synthesise cyclic oligoadenylate (cOA) upon detecting foreign RNA, activating ancillary nucleases that can be toxic to cells, necessitating mechanisms to remove cOA in systems that operate via immunity rather than abortive infection. Previously, we demonstrated that the *Sulfolobus solfataricus* type III-D CRISPR complex generates cyclic tetra-adenylate ($cA_4$), activating the ribonuclease Csx1, and showed that subsequent RNA cleavage and dissociation acts as an 'off-switch' for the cyclase activity. Subsequently, we identified the cellular ring nuclease Crn1, which slowly degrades $cA_4$ to reset the system (Rouillon et al., 2018), and demonstrated that viruses can subvert type III CRISPR immunity by means of a potent anti-CRISPR ring nuclease variant AcrIII-1. Here, we present a comprehensive analysis of the dynamic interplay between these enzymes, governing cyclic nucleotide levels and infection outcomes in virus-host conflict.

*For correspondence:
mfw2@st-and.ac.uk

Present address: [†]Department of Molecular Sensory Systems, Center of Advanced European Studies and Research (Stiftung Caesar), Bonn, Germany

**Competing interests:** The authors declare that no competing interests exist.

## Introduction

CRISPR systems are widespread in archaea and bacteria, providing adaptive immunity against invading mobile genetic elements (MGE) (*Sorek et al., 2013*; *Makarova et al., 2020*). Type III CRISPR systems (*Figure 1*) are multi-functional effector proteins that have specialised in the detection of foreign RNA (*Tamulaitis et al., 2017*; *Zhu et al., 2018*). The large subunit, Cas10, harbours two enzyme active sites that are activated by target RNA binding: a DNA-cleaving HD nuclease domain (*Samai et al., 2015*; *Elmore et al., 2016*; *Estrella et al., 2016*; *Kazlauskiene et al., 2016*) and a cyclase domain for cyclic oligoadenylate (cOA) synthesis (*Kazlauskiene et al., 2017*; *Niewoehner et al., 2017*; *Rouillon et al., 2018*). The third enzymatic activity of type III systems is situated in the Cas7 subunit of the complex, which cleaves bound RNA targets and in turn regulates Cas10 enzymatic activities (*Tamulaitis et al., 2014*; *Rouillon et al., 2018*; *Johnson et al., 2019*; *Nasef et al., 2019*). The cyclase domain polymerises ATP into cOA species consisting of between 3 and 6 AMP subunits (denoted $cA_3$, $cA_4$ etc.), in varying proportions (*Kazlauskiene et al., 2017*; *Niewoehner et al., 2017*; *Rouillon et al., 2018*; *Grüschow et al., 2019*; *Nasef et al., 2019*). cOA second messengers activate CRISPR ancillary nucleases of the Csx1/Csm6, Can1 (CRISPR associated nuclease 1) and NucC families, which drive the immune response against MGEs (*Kazlauskiene et al., 2017*; *Niewoehner et al., 2017*; *Rouillon et al., 2018*; *Grüschow et al., 2019*; *Lau et al., 2020*; *McMahon et al., 2020*). To date, $cA_4$ appears to be the most widely used signalling molecule by type III CRISPR systems (*Grüschow et al., 2019*). The ribonuclease activity of Csx1/Csm6 is crucial for the clearance of MGEs (*Hatoum-Aslan et al., 2014*; *Foster et al., 2019*; *Grüschow et al., 2019*), particularly when viral genes are transcribed late in infection, at low levels or mutated (*Hatoum-Aslan et al., 2014*; *Jiang et al., 2016*; *Rostøl and Marraffini, 2019*).

In our previous study, we demonstrated that the type III-D system from *Sulfolobus solfataricus* synthesises predominantly $cA_4$, which activates the CRISPR ancillary ribonuclease Csx1. We examined the first regulatory step in cOA synthesis in detail and demonstrated that target RNA cleavage and dissociation from the complex shut-off cOA synthesis (*Rouillon et al., 2018*). Since CRISPR ancillary nucleases degrade nucleic acids non-specifically, cellular as well as viral targets are destroyed. Collateral cleavage of self-transcripts by a Csm6 enzyme has previously been shown to result in cell growth arrest (*Rostøl and Marraffini, 2019*). Therefore, in addition to regulating the synthesis of cOA, cells need a mechanism to remove extant cOA if they are to return to normal growth. To solve this problem, *S. solfataricus* encodes CRISPR-associated ring nuclease 1 (Crn1) family enzymes (*Athukoralage et al., 2018*). Crn1 enzymes slowly degrade $cA_4$ to yield di-adenylate products incapable of activating Csx1. In other species Csm6 proteins have evolved catalytic CARF domains

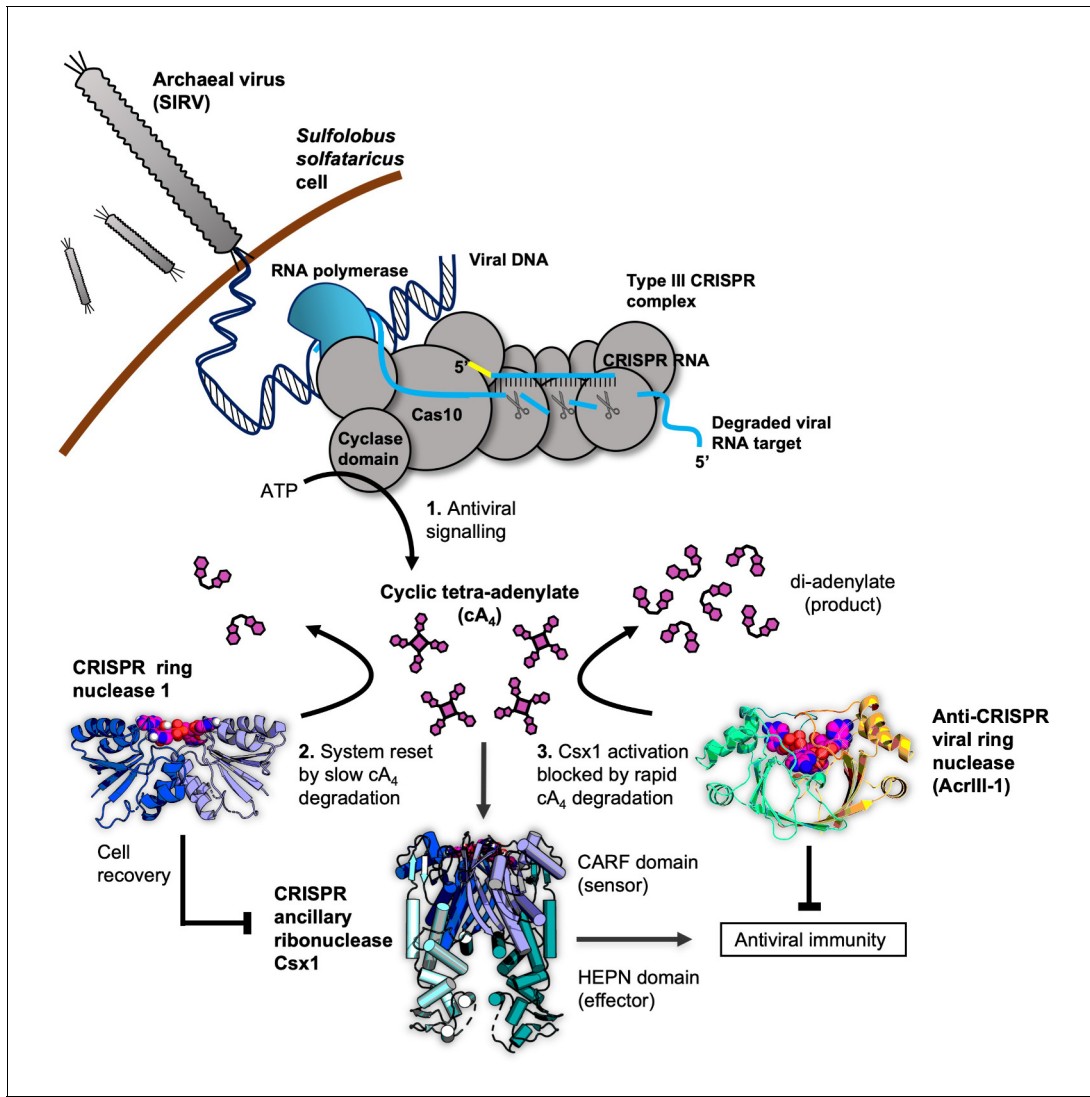

**Figure 1.** Cartoon of type III CRISPR cyclic nucleotide signalling and defence in *Sulfolobus solfataricus*. The Cas10 subunit of the type III CRISPR complex synthesises cyclic tetra-adenylate ($cA_4$) when viral RNA transcripts are detected. Target RNA cleavage shuts-off $cA_4$ synthesis. $cA_4$ binds to the CARF (CRISPR associated Rossmann Fold) domain of CRISPR ancillary nuclease Csx1 and allosterically activates its HEPN (Higher Eukaryotes and Prokaryotes Nucleotide binding) domain, which degrades RNA non-specifically within the cell. Extant $cA_4$ is degraded slowly by CRISPR ring nucleases (Crn1 family) which likely facilitate cell recovery after clearing the virus. Viral anti-CRISPR ring nucleases (AcrIII-1 family) degrade $cA_4$ rapidly to stop activation of ancillary defence enzymes such as Csx1 and supress antiviral immunity.

capable of degrading $cA_4$, thereby acting as their own 'off-switches' to their RNase activity (*Athukoralage et al., 2019*; *Jia et al., 2019*). Unsurprisingly, archaeal viruses and bacteriophage have co-opted this regulatory strategy in order to subvert type III CRISPR defence. Many archaeal viruses and bacteriophage encode a ring nuclease anti-CRISPR (AcrIII-1), unrelated to Crn1, which neutralises the type III response by rapidly degrading $cA_4$ to prevent ancillary nuclease activation (*Athukoralage et al., 2020*).

It is clear that the $cA_4$ antiviral second messenger is at the centre of a network of interactions that are crucial for effective type III CRISPR defence against MGE. Here, we show that detection of even a single molecule of invading RNA has the potential to generate a large signal amplification via synthesis of $cA_4$ that in turn activates the non-specific degradative ribonuclease Csx1. We explore how a cellular ring nuclease can return the cell to a basal state and how viruses can subvert the system. By quantifying and modelling the equilibria and reactions that take place in the arena of type III CRISPR defence, we build a comprehensive model of this dynamic, life or death process.

## Results

While the control of cOA synthesis by target RNA binding and cleavage is now understood reasonably well, the full implications of cOA generation in a virally-infected cell are not. This requires a detailed knowledge of the levels of cOA produced, consequences for antiviral defence enzymes and the effects of cOA degrading enzymes from cellular and viral sources. These were the aims of our study.

### Signal amplification on $cA_4$ production

We first investigated the extent of signal amplification that occurs in a cell from detection of a single viral RNA and generation of the $cA_4$ second messenger. Using the *S. solfataricus* type III-D CRISPR effector, we varied the concentration of target RNA and quantified the resultant $cA_4$ production. As previously observed (*Rouillon et al., 2018*), increasing the target RNA concentration resulted in increased $cA_4$ production (*Figure 2*). Quantification of the concentration of $cA_4$ generated was accomplished by using $\alpha$-$^{32}$P-ATP and quantification of products using a phosphorimager in comparison to standards (*Figure 2—figure supplement 1*). We observed that approximately 1000 molecules (980 ± 24) of $cA_4$ were generated per molecule of RNA, over a range of 10–100 nM target RNA (*Figure 2*). When a poorly-cleavable target RNA species containing phosphorothioates was used as the substrate, the amount of $cA_4$ generated increased approximately 3-fold (3100 ± 750, *Figure 2*), confirming the important role of RNA cleavage for deactivation of the cyclase domain (*Rouillon et al., 2018*; *Nasef et al., 2019*). Additionally, analysis of previously published data enabled us to determine the catalytic rate constant ($k$) of $cA_4$ synthesis by the *S. solfataricus* type III effector as 0.04 ± 0.01 min$^{-1}$ at 70°C (*Figure 2—figure supplement 2*).

Given that *S. solfataricus* cells are cocci with a diameter of approximately 0.7 µm, the volume of an average cell can be calculated as approximately 0.3 fL (by comparison, *E. coli* has a cell volume of 1 fL [*Kubitschek and Friske, 1986*]). Using Avogadro's number, 1000 molecules equates to an intracellular concentration of 6 µM $cA_4$ in *S. solfataricus*. Thus, detection of one viral RNA in the cell could result in the synthesis of 6 µM $cA_4$, 10 RNAs – 60 µM, etc. The upper limits of $cA_4$ generation could be defined by the number of viral target RNAs present, the number of type III effectors carrying a crRNA matching that target, or even conceivably the amount of ATP available for $cA_4$ generation.

### Kinetic parameters of the Csx1 ribonuclease

The $cA_4$ second messenger binds to CARF family proteins to elicit an immune response. To understand the concentration of $cA_4$ required to activate an antiviral response, we determined the dissociation constant of the major ancillary ribonuclease Csx1 for the $cA_4$ activator. Using radioactive $cA_4$, we titrated an increasing concentration of Csx1 protein and subjected the mixture to native gel electrophoresis (*Figure 3A,B*). $cA_4$ was bound by Csx1 with a dissociation constant ($K_D$) of 130 ± 20 nM. Thus, even one viral target RNA detected by the type III CRISPR system should generate enough $cA_4$ (6 µM) to fully activate the Csx1 ribonuclease for defence. We proceeded to estimate the binding affinity of a ribonuclease-deficient Csx1 variant for its RNA target, yielding an apparent dissociation constant of approximately 5 µM (*Figure 3C*). Finally, we measured the initial velocity of RNA degradation at a variety of RNA concentrations under steady state conditions and determined the

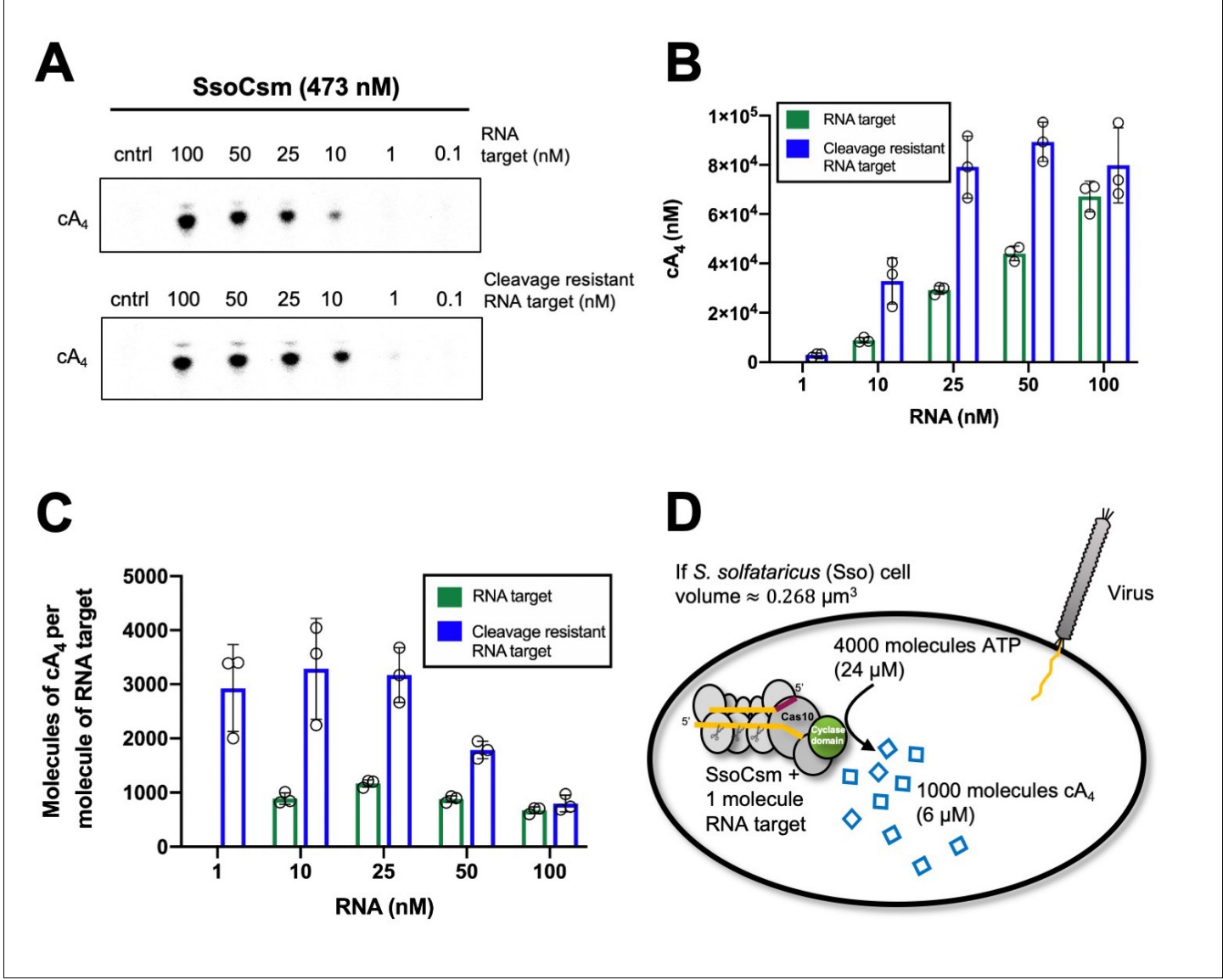

**Figure 2.** Approximately 1000 molecules $cA_4$ are made per molecule of RNA target. (**A**) Upper panel shows phosphorimages of thin-layer chromatography of cyclic tetra-adenylate ($cA_4$) made by *S. solfataricus* (Sso) Csm complex (470 nM carrying the CRISPR RNA A26) across a range of RNA target concentrations (0.1, 1, 10, 25,100 nM) complementary to the A26 CRISPR RNA at 70°C. Lower panel shows $cA_4$ synthesised with a cleavage resistant (phosphorothioate) form of the RNA target. (**B**) Bar graph of the concentration of $cA_4$ generated with increasing cleavable and cleavage-resistant RNA target generated by quantifying the densiometric signals from A, with an $\alpha$-$^{32}$P-ATP standard curve (*Figure 2—figure supplement 1*). Error bars indicate the standard deviation of the mean of three technical replicates, with individual data points shown as clear circles. No data are shown for 1 nM cleavable RNA target as $cA_4$ generated was below detection limits. (**C**) Bar chart quantifying the number of molecules of $cA_4$ generated per molecule of cleavable or cleavage resistant target RNA across a range of RNA target concentrations. On average SsoCsm synthesised 980 ± 24 and 3100 ± 750 molecules of $cA_4$ per molecule of cleavable and cleavage resistant target RNA, respectively. (**D**) Cartoon depicting the cellular implications of ~1000 molecules of $cA_4$ generated per molecule of RNA target, which in *S. solfataricus* would equate to ~6 µM $cA_4$ within the cell.

The online version of this article includes the following source data and figure supplement(s) for figure 2:

**Source data 1.** Excel spreadsheet with raw data.

**Figure supplement 1.** Example of ATP standard curve used to determine the concentration of ATP converted to cyclic tetra-adenylate ($cA_4$).

**Figure supplement 2.** Rate of $cA_4$ synthesis by *S. solfataricus* type III-D effector complex.

**Figure supplement 2—source data 1.** Excel speadsheet with raw data.

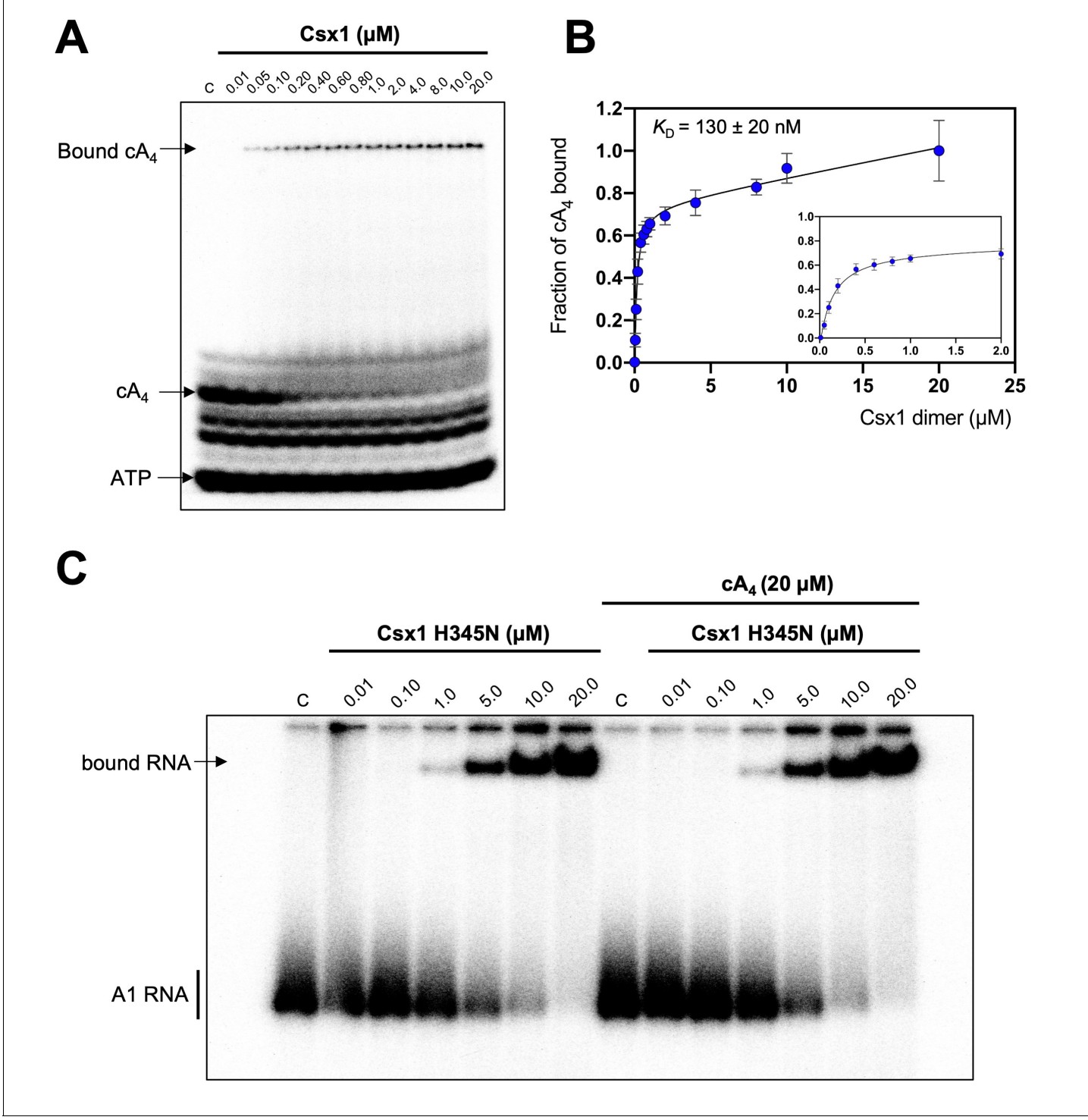

**Figure 3.** Csx1 binds cA$_4$ with high affinity and RNA with relatively low affinity. (**A**) Phosphorimage of native gel electrophoresis visualising cA$_4$ (20 nM) binding by Csx1 (concentrations as indicated in the figure). The other bands visible are due to unreacted ATP and other linear nucleotide products. (**B**) Plot of fraction of cA$_4$ bound by Csx1. Error bars indicate the standard deviation of the mean of four technical replicates and the data were fitted to a quadratic equation incorporating a term for non-specific binding. The inset plot is a magnification of cA$_4$ binding between 0.01 and 5 μM Csx1 dimer concentrations. (**C**) Phosphorimage of native gel electrophoresis visualising A1 substrate RNA binding by Csx1 H345N protein dimer in the absence (left hand-side) or presence (right hand-side) of unlabelled cA$_4$ (20 μM). The image shown is representative of three technical replicates. Control c – RNA alone.

The online version of this article includes the following source data for figure 3:

**Source data 1.** Csx1 binding to cA$_4$ and RNA.

multiple-turnover kinetic constant ($k_{cat}$) for cA$_4$-activated RNA cleavage by Csx1 as 0.44 ± 0.03 min$^{-1}$ at 70°C (*Figure 4*).

## Kinetic and equilibrium constants of the ring nucleases Crn1 and AcrIII-1

We have previously established that Crn1 cleaved cA$_4$ at a rate of 0.089 ± 0.003 min$^{-1}$ at 50°C, while AcrIII-1 cleaved cA$_4$ at a rate of 5.4 ± 0.38 min$^{-1}$, about 60-fold faster (*Athukoralage et al., 2020*). The difference in reaction rates probably reflects the different roles of the two enzymes, with Crn1 working in conjunction with the type III CRISPR defence and AcrIII-1 opposing it. To quantify the interaction between ring nucleases and cA$_4$, we titrated radioactively labelled cA$_4$ with either Crn1 or AcrIII-1 and visualised cA$_4$ binding by phosphorimaging following native gel electrophoresis. Crn1 bound cA$_4$ with an apparent $K_D$ of ~50 nM, while the inactive H47A variant of AcrIII-1 bound cA$_4$ with an apparent $K_D$ of ~25 nM (*Figure 5*). Thus, both ring nucleases bound cA$_4$ three to four-fold more tightly than Csx1.

## Kinetic modelling of the antiviral signalling pathway and its regulation by cA$_4$ degrading enzymes

We entered the experimentally determined kinetic and equilibria parameters into the KinTek Global Kinetic Explorer software package and generated a model to simulate RNA degradation by Csx1 and the effects of ring nucleases over time (*Figure 6A*). Enzyme concentrations were set initially at 1 µM, based on published studies of transcript levels (*Ortmann et al., 2008*; *Wurtzel et al., 2010*), but were varied during modelling to assess the influence of enzyme concentration on RNA cleavage. We standardised the reaction temperature at 70°C, close to the growth temperature of *S. solfataricus*. This necessitated an estimation of the rate constants of Crn1 and AcrIII-1, which were measured at lower temperatures, based on a 2-fold increase in activity for each 10°C rise in temperature, in line

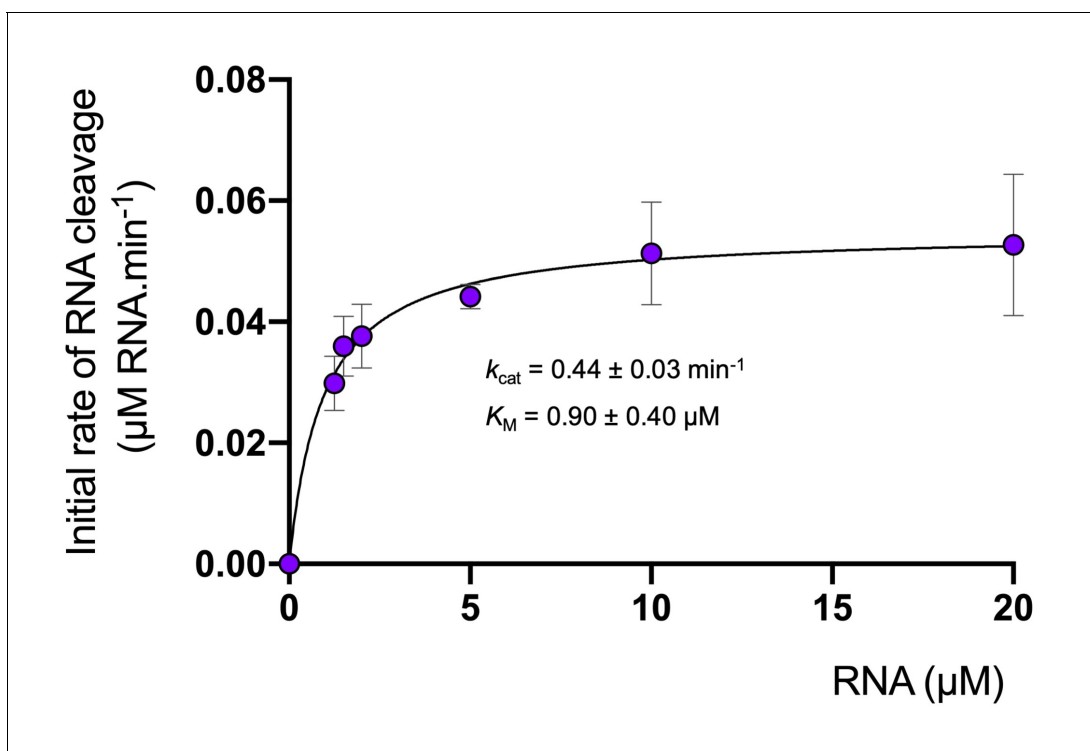

**Figure 4.** Degradation of RNA by Csx1. Analysis of multiple-turnover, steady state kinetics of A1 RNA cleavage by Csx1 (125 nM dimer) at 70°C. The data were fitted to the Michaelis-Menten equation and error bars show the standard deviation of the mean of three technical replicates.

The online version of this article includes the following source data for figure 4:

**Source data 1.** Excel spreadsheet with data for kinetics of Csx1.

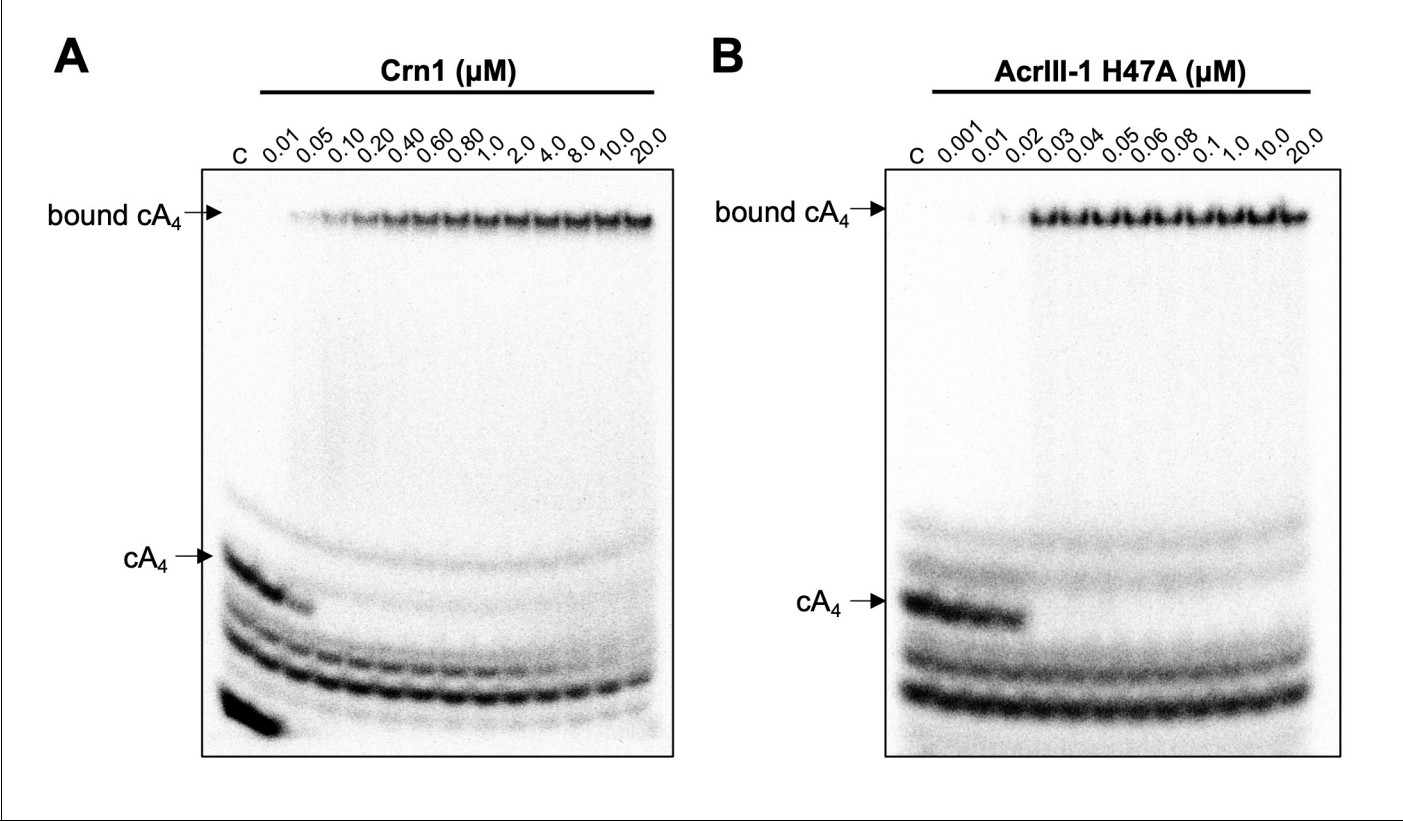

**Figure 5.** Crn1 and AcrIII-1 bind $cA_4$ with high affinity. Phosphorimages of native gel electrophoresis visualising radiolabelled cyclic oligoadenylate (cOA) binding by (A) Crn1 (B) and catalytically inactive AcrIII-1 (SIRV1 gp29 H47A). Crn1 binds $cA_4$ (10 nM) with an apparent dissociation constant ($K_D$) of ~50 nM, whereas AcrIII-1 binds $cA_4$ with an apparent $K_D$ of ~25 nM. The images shown are representative of three technical replicates. Control c – cOA alone. The other bands near the bottom of the gel are caused by unreacted ATP and other linear products.

with our previous studies of these enzymes (*Athukoralage et al., 2018*; *Athukoralage et al., 2020*). To model the generation of $cA_4$ by the Csm:target RNA complex, we set the concentration of that complex at 6, 60 or 600 µM (equivalent to low, medium and high levels of infection), allowing the generation of the equivalent concentration of $cA_4$ with the measured rate constant of 0.04 min$^{-1}$ (*Figure 6A*). Henceforth, this will be referred to as 6, 60 or 600 µM $cA_4$ for simplicity.

In the absence of any ring nuclease activity, 1 µM Csx1 was fully activated, degrading all 1000 µM RNA provided in the simulation within 2400 min, regardless of the simulated level of infection (*Figure 6—figure supplement 1*). The lack of titration of Csx1 activity observed here is due to its high affinity for $cA_4$, resulting in full activation even at very low simulated infection levels. We observed a rapid initial increase in the levels of $cA_4$ due to synthesis by the activated Csm complex (*Figure 6B*). In the simulations, 1 µM Crn1 degraded 60 µM $cA_4$ within 120 min, but took over 1200 min to degrade 600 µM $cA_4$ (*Figure 6B*). These differences were reflected in the levels of Csx1-mediated RNA degradation, which were much higher for 600 µM $cA_4$ than for the lower concentrations (*Figure 6C*). Thus, the kinetic modelling demonstrated that addition of a ring nuclease activity allows the cell to respond differently to varied levels of infection, and therefore $cA_4$ concentration, resulting in a range of RNA degradation levels.

We next introduced the AcrIII-1 enzyme to the simulation, to model the effect of the Acr on the host defence system. Strikingly, when AcrIII-1 was present at 1 µM, even 600 µM $cA_4$ was degraded rapidly (the bulk within 10 min). As a result, Csx1 activity and therefore RNA degradation were strongly suppressed (*Figure 6D,E*). The long tail of $cA_4$ observed in these simulations represents $cA_4$ bound to the Csx1 enzyme and thus unavailable for degradation by AcrIII-1, reflected in the persistence of active Csx1-$cA_4$ complex in the simulations over an extended period (*Figure 6—figure supplement 2*). Notably, at the highest simulated infection level of 600 µM $cA_4$, Csx1 persists in a 100%

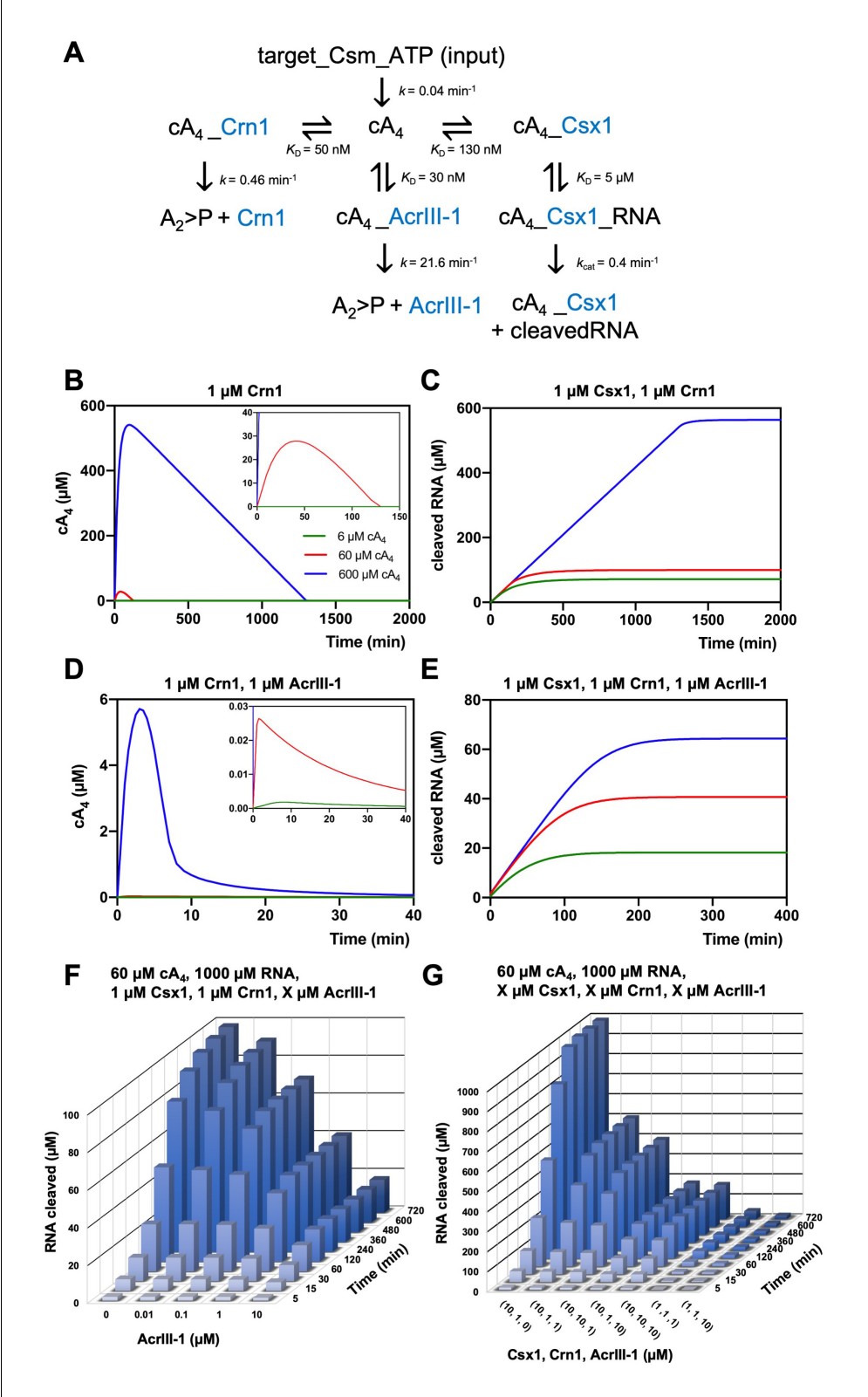

**Figure 6.** Modelling of *S. solfataricus* antiviral signalling. (**A**) Schematic showing kinetic and equilibrium parameters inserted into the KinTek Global Kinetic Explorer software for modelling the type III CRISPR defence illustrated in *Figure 1*. Parameters have been determined in this study with the following exceptions: Crn1 rate constant of 0.46 min$^{-1}$ at 70°C estimated from rate of 0.23 min$^{-1}$ measured at 60°C and AcrIII-1 rate constant of 21.6 min$^{-1}$ at 70°C estimated from rate of 5.4 min$^{-1}$ measured at 50°C (*Athukoralage et al., 2020*). The parameter 'target-Csm-ATP' was set at 6, 60 or 600

*Figure 6 continued on next page*

*Figure 6 continued*

μM in simulations. Underscores connecting two variables indicate their relationship in a complex. cA$_4$, cyclic tetra-adenylate; Crn1, CRISPR ring nuclease 1; AcrIII-1, viral ring nuclease anti-CRISPR SIRV1 gp29; Csx1, CRISPR ancillary ribonuclease; A$_2$ >P, di-adenylate containing 2',3' cyclic phosphate (product of cA$_4$ cleavage). (B) Free cA$_4$ (600 μM, blue; 60 μM, red; 6 μM, green) concentration and (C), RNA cleavage in presence of 1 μM Crn1 and 1 μM Csx1. (D) and (E) show equivalent plots in the presence of 1 μM AcrIII-1. Insets where present show a magnified view of the start of each plot. (F) 3D plot visualising concentration of RNA (1000 μM at start) cleaved by Csx1 in response to 60 μM cA$_4$ made by Csm complex, 1 μM Crn1 and varying amounts of AcrIII-1 across a range of doubling endpoints. (G) 3D plot visualising concentration of RNA (1000 μM at start) cleaved by Csx1 in response to 60 μM cA$_4$ made by Csm complex, and varying concentrations of Csx1, Crn1 and AcrIII-1 enzymes.

The online version of this article includes the following source data, source code and figure supplement(s) for figure 6:

**Source data 1.** Raw data from modelling.
**Source data 2.** Extracted data from modelling.
**Source code 1.** KinTek model code for kinetic modelling in *Figure 6*.
**Figure supplement 1.** Modelling of RNA cleavage by Csx1 stimulated by cA$_4$ synthesised in response to varying target RNA concentrations.
**Figure supplement 2.** Modelling of the relationship between Csx1 RNA cleavage and deactivation by ring nucleases with varying cA$_4$ concentrations.
**Figure supplement 3.** Modelling of the effect of varying AcrIII-1 concentration on RNA cleaved by Csx1 in response to cA$_4$ generated by 600 μM target RNA.

activated form for over 1200 min in the absence of AcrIII-1, but is reduced to 50% activated within 150 min in its presence, with similar effects seen at lower cA$_4$ concentrations (*Figure 6—figure supplement 2*).

The concentration of AcrIII-1 within *S. solfataricus* cells during infection is not known. We therefore varied AcrIII-1 concentration in the model (with 60 μM cA$_4$, 1 μM Csx1 and Crn1) and simulated RNA cleavage (*Figure 6F*). AcrIII-1 concentrations as low as 100 nM reduced RNA cleavage by about 30%, and higher levels reduced RNA cleavage by Csx1 markedly (*Figure 6F*). This relationship held in simulations with 600 μM cA$_4$ (*Figure 6F*, *Figure 6—figure supplement 3*). During infection, a positive correlation between AcrIII-1 concentration and viral transcript levels would be required for continued escape from Csx1-mediated CRISPR defence – a reasonable assumption. Finally, we simulated the effects of increasing Csx1 on RNA cleavage in our infection model. Increasing Csx1 concentration 10-fold led to a significantly more robust immune response (reflected in high levels of RNA degradation) (*Figure 6G*) that could not be fully overcome by increasing the concentrations of Crn1 or AcrIII-1 in proportion (compare for example 10,10,10 with 1,1,1 μM in *Figure 6G*). Hence, increasing the expression levels of defence nucleases such as Csx1 in response to viral infection could provide an effective means for cells to combat viruses armed with Acrs such as AcrIII-1, but perhaps with the cost of increased toxicity.

## Discussion

### Signal amplification in type III CRISPR defence

In this study, we used biochemical data to build a kinetic model of the type III CRISPR antiviral signalling pathway within *S. solfataricus* cells and examined the capacity of CRISPR and anti-CRISPR ring nucleases for its regulation. Quantification of cA$_4$ generated by the SsoCsm complex in vitro revealed that ~ 1000 molecules of cA$_4$ are made per RNA target, amounting to a concentration of 6 μM in the cell if replicated in vivo. This large degree of signal amplification would ensure that detection of 1 RNA target could generate sufficient amounts of cA$_4$ to fully activate the ribonuclease effector protein Csx1, which has a dissociation constant for cA$_4$ of 0.13 μM. Given the large signal amplification observed here, it seems likely that some means of cOA degradation, either via self-limiting ribonucleases (*Athukoralage et al., 2019*; *Jia et al., 2019*) or dedicated ring nucleases (*Athukoralage et al., 2018*), will be essential for type III CRISPR systems to provide immunity rather than elicit abortive infection. Indeed, growth arrest has been observed for cOA activated Csm6 during bacteriophage infection (*Rostøl and Marraffini, 2019*). This life or death decision in response to genotoxic stress has also been observed in *S. islandicus*, which becomes dormant upon viral infection and subsequently dies if virus remains in culture (*Bautista et al., 2015*). In recent years, diverse CRISPR systems have been implicated in abortive infection or cell dormancy. The Type I-F CRISPR system of *Pectobacterium atrosepticum* was found to provide population protection by aborting infection when infected by virulent phage (*Watson et al., 2019*). Likewise, the in-trans collateral

RNA cleavage of *Listeria seeligeri* Cas13a resulted in cell dormancy, providing herd immunity to the bacterial population (*Meeske et al., 2019*). Similarly, in ecological contexts, it is possible that different multiplicities of viral infection illicit different outcomes from the type III CRISPR response that benefit either the individual cell or the population.

## Cellular and viral ring nucleases reset the system in fundamentally different ways

Biochemical comparison of Crn1 and AcrIII-1 revealed that both enzymes bind $cA_4$ with dissociation constants around 40 nM, around 3-fold tighter than observed for Csx1. However, Crn1 is a much slower enzyme. Kinetic modelling of the antiviral signalling pathway confirms that Crn1 is effective only at low levels of viral gene expression, where it has the potential to neutralise the toxicity associated with $cA_4$-activated ribonucleases to offer a route for cell recovery without abrogating immunity. In contrast, the much faster reaction kinetics of the anti-CRISPR ring nuclease means it can rapidly deactivate Csx1 and immunosuppress cells even under very high RNA target (and thus $cA_4$) levels.

Our modelling suggests that the rapid turnover of $cA_4$ by AcrIII-1 over a wide concentration range greatly limits RNA cleavage by deactivating defence enzymes. Therefore, the deployment of AcrIII-1 upon viral infection may not only promote viral propagation but also safeguard cellular integrity until viral release by lysis. Recent studies have uncovered that sequentially infecting phage evade CRISPR defences by exploiting the immunosuppression achieved by Acr enzymes from failed infections (*Borges et al., 2018*; *Landsberger et al., 2018*). Further, these immunosuppressed cells have been shown to be susceptible to Acr-negative phage infections, highlighting the complex ecological consequences of supressing CRISPR immunity (*Chevallereau et al., 2020*). In *Sulfolobus* Turreted Icosahedral virus (STIV), the AcrIII-1 gene *B116* is expressed early in the viral life cycle (*Ortmann et al., 2008*). Therefore, AcrIII-1 accumulation in the cell, possibly from early expression by unsuccessful viruses may, as our models demonstrate, favour the success of latter viral infections. Type III CRISPR systems also conditionally tolerate prophage (*Goldberg et al., 2014*), and unsurprisingly, AcrIII-1 is found in a number of prophages and integrative and conjugative elements. In these cases, constitutively expressed AcrIII-1 may further immunocompromise cells, and sensitise them to infection by phage otherwise eradicated by type III CRISPR defence. In the ongoing virus-host conflict, while increasing Csx1 concentration may allow better immunity when faced with AcrIII-1, upregulating AcrIII-1 expression in cells will undoubtedly offer viruses an avenue for counter offence.

It should be noted that the type III CRISPR locus of *S. solfataricus* contains a number of CARF domain proteins and their contribution to immunity has not yet been studied. In particular, the CARF-family putative transcription factor Csa3 appears to be involved in transcriptional regulation of CRISPR loci, including the adaptation and type I-A effector genes, when activated by $cA_4$ (*Liu et al., 2015*; *Liu et al., 2017*). These observations suggest that the $cA_4$ signal may transcend type III CRISPR defence in some cell types by activating multiple defence systems. However, by degrading the second messenger, AcrIII-1 has the potential to neutralise all of these.

## Cyclic nucleotides in prokaryotic defence systems

Cyclic nucleotide-based defence systems are emerging as powerful cellular sentinels against parasitic elements in prokaryotes. Mirroring the role of cyclic GMP-AMP synthase (cGAS) in eukaryotic defence against viruses as part of the cGAS-STING pathway, bacterial cGAS enzymes have recently been discovered that abort infection by activating phospholipases through cGAMP signaling (*Cohen et al., 2019*). Termed the cyclic-oligonucleotide-based antiphage signaling system (CBASS), a large number of additional cOA sensing effector proteins associated with CBASS loci remain uncharacterised, highlighting great diversity in the cellular arsenal used for defence (*Burroughs et al., 2015*; *Cohen et al., 2019*). Furthermore, diverse nucleotide cyclases have been identified that generate a range of cyclic nucleotides including cUMP-AMP, c-di-UMP and cAAG, which are also likely to function in novel antiviral signal transduction pathways (*Whiteley et al., 2019*). Type III systems also generate cyclic tri-adenylate ($cA_3$) and cyclic penta-adenylate ($cA_5$) molecules. Whereas no signalling role has yet been ascribed to $cA_5$, $cA_3$ has been demonstrated to activate a family of DNases termed NucC which abort infection by degrading the host genome prior to completion of the phage replication cycle (*Lau et al., 2020*).

The balance between immunity, abortive infection and successful pathogen replication is likely to be governed by enzymes that synthesise and degrade these cyclic nucleotide second messengers. Just as prokaryotes with type III CRISPR require a means to degrade cOA in appropriate circumstances, eukaryotic cells have enzymes that degrade cGAMP to regulate cGAS-STING mediated immunity (*Li et al., 2014*). Likewise, while prokaryotic viruses utilise AcrIII-1 to rapidly degrade $cA_4$, eukaryotic poxviruses utilise Poxins to subvert host immunity by destroying cGAMP (*Eaglesham et al., 2019*), and pathogenic Group B *Streptococci* degrade host c-di-AMP using the CndP enzyme to circumvent innate immunity (*Andrade et al., 2016*). The rate of discovery of new defence pathways and cyclic nucleotide signals is breath-taking. Analysis of the dynamic interplay between enzymes that leads to fluctuations in the levels of these second messengers is therefore of crucial importance if we are to achieve an understanding of these processes.

# Materials and methods

## Key resources table

| Reagent type (species) or resource | Designation | Source or reference | Identifiers | Additional information |
|---|---|---|---|---|
| Gene (*Sulfolobus solfataricus*) | Csm complex (eight subunits) | PMID:24119402 | | virus expression construct |
| Gene (*Sulfolobus solfataricus*) | Csx1 | PMID:29963983 | UniProtKB - Q97YD5 | plasmid expression construct |
| Gene (*Sulfolobus solfataricus*) | Crn1 | PMID:30232454 | UniProtKB - Q7LYJ6 | plasmid expression construct |
| Gene (*Sulfolobus islandicus rod-shaped virus 1*) | AcrIII-1 | PMID:31942067 | UniProtKB - Q8QL27 | plasmid expression construct |
| Software, algorithm | KinTek Kinetic Explorer | PMID:19897109 | | model constructed for this paper |

## Cyclic oligoadenylate (cOA) synthesis and visualisation

Cyclic tetra-adenylate ($cA_4$) made per RNA target (0.01, 0.1, 1, 10, 25 or 50 nM) was investigated in a 20 µl reaction volume incubating A26 RNA target or A26 phosphorothioate RNA target (*Table 1*) with 13.5 µg *Sulfolobus solfataricus* (Sso)Csm complex (~470 nM carrying A26 CRISPR RNA) in Csx1 buffer containing 20 mM MES pH 5.5, 100 mM K-glutamate, 1 mM DTT and 3 units SUPERase•In Inhibitor supplemented with 1 mM ATP, 5 nM $\alpha$-$^{32}$P-ATP and 2 mM $MgCl_2$ at 70°C for 2 hr. All samples were deproteinised by phenol-chloroform extraction (Ambion) followed by chloroform (Sigma-Aldrich) extraction prior to separating the cOA products by thin-layer chromatography (TLC). TLC was carried out as previously described (*Rouillon et al., 2019*). In brief, 1 µl of radiolabelled cOA product was spotted 1 cm from the bottom of a 20 × 20 cm silica gel TLC plate (Supelco Sigma-Aldrich). The TLC plate was placed in a sealed glass chamber pre-warmed at 37°C containing 0.5 cm of a running buffer composed of 30% $H_2O$, 70% ethanol and 0.2 M ammonium bicarbonate, pH 9.2. After TLC the plate was air dried and sample migration visualised by phosphor imaging. For analysis,

**Table 1.** Oligonucleotides.
CRISPR RNA A26 is shown 3' to 5'. Phosphorothioate linkages are indicated with an asterisk. Regions complementary to CRISPR RNA A26 are italicized.

| Crispr rna a26 | 3'-GCAACAATTCTTGCTGCAACAATCTTCAACCCATACCAGAAAGUUA |
|---|---|
| **Name** | **Sequence (5'—3')** |
| Target RNA A26 | AGGGUCGUUGUUAAGAACGACGUUGUUAGAAGUUGGGUAUGGUGGAGA |
| Phosphorothioate target RNA A26 | AGGGUCGUUGUUAAGAACGACGUUGU*U*A*GAAGUUGGGU*A*U*GGUGGAGA |
| A1 substrate RNA | AGGGUAUUAUUUGUUUGUUUCUUCUAAACUAUAAGCUAGUUCUCGGAGA |

densiometric signals corresponding to cA$_4$ was quantified as previously described (*Rouillon et al., 2019*).

## Generation of α-$^{32}$P-ATP standard curves

cA$_4$ synthesis was visualised by incorporation of 5 nM α-$^{32}$P-ATP added together with 0.5 mM ATP at the start of the reaction. Therefore, to calculate the concentration of ATP used for cA$_4$ synthesis, α-$^{32}$P-ATP standard curves were generated in duplicate, starting with 5 nM α-$^{32}$P-ATP within a 20 µl volume to represent the densiometric signal corresponding to the complete conversion of 0.5 mM ATP into cOA. Serial two-fold dilutions of 5 nM α-$^{32}$P-ATP and 0.5 mM ATP starting from a 20 µl volume were made and 1 µl of each dilution was spotted on a silica plate and phosphorimaged alongside TLC separating cOA made with varying RNA target concentrations. After phosphorimaging, the densiometric signals of the serial dilutions were quantified, averaged and plotted against ATP concentration starting from 0.5 mM and halving with each two-fold dilution. A line of best fit was then drawn. The concentration of ATP used to synthesise cA$_4$ was calculated by entering the densiometric signal of the cA$_4$ product into to equation of the line of best fit for the α-$^{32}$P-ATP standard curve. The concentration of cA$_4$ generated was derived by dividing the concentration of ATP incorporated by four to account for polymerisation of four ATP molecules to generate one molecule of cA$_4$. Finally, the molecules of cA$_4$ made per RNA was calculated by dividing the cA$_4$ concentration generated by the concentration A26 RNA target used for cOA synthesis.

Calculation determining the concentration of cA$_4$ made when one RNA target is detected within a *S. solfataricus* cell of ≈ 0.8 µm (0.6–1.0 µm) diameter.

$$\text{Volume } (V) = \frac{4}{3}\pi r^3 \text{ and } r = \frac{1}{2}d$$

$$r = \frac{1}{2} \times 0.8\,\mu m$$

$$r = 0.4\,\mu m$$

$$V = \frac{4}{3}\pi \times (0.4\,\mu m)^3$$

$$V = 0.268\,\mu m^3$$

$$1\,\mu m^3 = 1\,fL$$

$$0.268\,\mu m^3 = 0.268\,fL = 2.68 \times 10^{-13}\,mL$$

$$1 \text{ mole of RNA } = 6.022 \times 10^{23} \text{ molecules of RNA}$$

$$1 \text{ molecule of RNA } = 1 \div 6.022 \times 10^{23} = 1.661 \times 10^{-24} \text{ moles of RNA}$$

As ~1000 molecules of cA$_4$ is made per 1 molecule of RNA

$$1.661 \times 10^{-24} \text{ moles } \times 1000 = 1.661 \times 10^{-21} \text{ moles of cA}_4$$

Concentration (M) = moles / Volume (L)

$$1.661 \times 10^{-21} \text{ moles } \div 2.68 \times 10^{-16}\,L = 6.20 \times 10^{-6}\,M \text{ or } 6.20\,\mu M \text{ cA}_4$$

## Electrophoretic mobility shift assays to determine cA$_4$ equilibrium binding constants

~20 nM radioactively-labelled cA$_4$ generated using the SsoCsm was incubated with increasing concentrations of Csx1 (0.01, 0.05, 0.10, 0.20, 0.40, 0.60, 0.80, 1,0, 2.0, 4.0, 8.0, 10.0, 20.0 µM protein

dimer) in buffer containing 20 mM Tris-HCl pH 7.5, 150 mM NaCl, 2 mM $MgCl_2$ supplemented with 2 µM Ultrapure Bovine Serum Albumin (Invitrogen) for 10 min at 25˚C. A reaction volume equivalent of 20% (v/v) glycerol was then added prior to loading the samples on a 15% polyacrylamide, 1 X TBE gel. Electrophoresis was carried out at 28˚C and 250 V. Gels were phosphor imaged overnight at −80˚C. For investigating RNA binding, 50 nM 5'-end radiolabelled and gel purified A1 RNA was incubated with Csx1 variant H345N (0.01, 0.10, 1.0, 5.0, 10.0, 20.0 µM protein dimer) in the presence or absence of 20 µM $cA_4$ for 15 min at 40˚C. To examine $cA_4$ binding by Crn1,~10 nM radiolabelled SsoCsm $cA_4$ was incubated with Sso2081 (0.01, 0.05, 0.10, 0.20, 0.40, 0.60, 0.80, 1,0, 2.0, 4.0, 8.0, 10.0, 20.0 µM protein dimer) on ice for 15 min before gel electrophoresis as described above but at 300V and at 4˚C. $cA_4$ binding by AcrIII-1 was examined by incubating ~10 nM radiolabelled SsoCsm $cA_4$ with SIRV1 gp49 H47A (0.001, 0.01, 0.02, 0.03, 0.04, 0.05, 0.06, 0.08, 0.10, 1.0, 10.0, 20.0 µM protein dimer) for 10 min at 25˚C before gel electrophoresis at 30˚C as described above. For analysis densiometric signal corresponding to $cA_4$ bound protein was quantified. The densiometric count corresponding to $cA_4$ bound to 20 µM Csx1 dimer was used to represent 100% binding and densiometric counts from other lanes were normalised to this value within each replicate. Error of the 100% bound (20 µM Csx1 dimer) densiometric count was derived by calculating the area adjusted count for each replicate and then the standard deviation of their mean, reporting the standard deviation as a fraction of the mean set as 100% bound. The data were fitted to a quadratic equation with an adjust for nonspecific binding ($Y = (ymax-ymin)*((x + [ligand] + Kd) - ((x+[ligand]+Kd)2–4*x*[ligand])^0.5)/(2*[ligand]) +a*x+b$) using GraphPad Prism 8.

## Multiple turnover kinetics of RNA cleavage by Csx1

Multiple turnover kinetic experiments were carried out by incubating Csx1 (0.125 µM dimer) with radiolabelled (50 nM) and unlabelled RNA A1 to a final concentration of 1.25, 1.5, 2.0, 5.0, 10.0, or 20.0 µM RNA in Csx1 buffer at 70˚C. Control reactions with no protein and with protein and RNA in the absence of $cA_4$ were included. 10 µl reaction aliquots were quenched by adding to phenol-chloroform and vortexing and the different time-points at which reactions were quenched for each RNA concentration is indicated in data transparency. Deproteinised products were run on a 7 M urea, 20% acrylamide, 1 X TBE gel at 45˚C as previously described (*Rouillon et al., 2019*), and phosphor-imaged overnight at −80˚C. Each experiment was carried out in triplicate. Fraction of RNA cut was plotted against time and a linear regression was carried out on the linear portion of each plot to determine the initial rate of RNA cleavage at each RNA concentration. To determine the parameters $k_{cat}$ and $K_M$, the average initial rate of RNA cleavage was plotted against RNA concentration and fitted to the Michaelis-Menten equation using GraphPad Prism 8.

## Modelling antiviral signalling and its control by ring nucleases

Modelling was carried out using the KinTek Explorer eight software package (*Johnson, 2009*), which is available from (https://kintekcorp.com/software). Experiments were modelled and simulated using kinetic and equilibrium paramters detemined experimentally as described in *Figure 6A*. The following steps were inserted to generate the model:

    target_Csm_ATP = target_Csm + cA4 (irreversible)
    cA4 + Csx1 = cA4_Csx1
    cA4_Csx1 + RNA = cA4_Csx1_RNA
    cA4_Csx1_RNA = cA4_Csx1_cleavedRNA (irreversible)
    cA4_Csx1_cleavedRNA = cA4_Csx1 + cleavedRNA
    cA4 + Crn1 = cA4_Crn1
    cA4_Crn1 = A2 + Crn1 (irreversible)
    cA4 + Vrn = cA4_Vrn cA4_Vrn = A2 + Vrn (irreversible)

Simulations were carried out varying target_Csm_ATP concentration (6, 60 and 600 µM) while Csx1, Crn1 (Sso2081) and AcrIII-1 concentrations were fixed at 1 µM dimer, or varied depending on the simulation, with total substrate RNA in the cell fixed at 1000 µM.

## Acknowledgements

This work was supported by a grant from the Biotechnology and Biological Sciences Research Council (Grant REF BB/S000313/1 to MFW) and the Wellcome Trust (Grant 210486/Z/18/Z to CMC)

## Additional information

### Funding

| Funder | Grant reference number | Author |
| --- | --- | --- |
| Biotechnology and Biological Sciences Research Council | BB/S000313/1 | Malcolm F White |
| Wellcome | 210486/Z/18/Z | Clarissa M Czekster |

The funders had no role in study design, data collection and interpretation, or the decision to submit the work for publication.

### Author contributions

Januka S Athukoralage, Formal analysis, Investigation, Visualization, Methodology, Writing - original draft, Writing - review and editing; Shirley Graham, Christophe Rouillon, Sabine Grüschow, Methodology, Writing - review and editing; Clarissa M Czekster, Visualization, Methodology, Writing - review and editing; Malcolm F White, Conceptualization, Formal analysis, Supervision, Funding acquisition, Writing - original draft, Project administration, Writing - review and editing

### Author ORCIDs

Januka S Athukoralage (iD) https://orcid.org/0000-0002-1666-0180
Shirley Graham (iD) http://orcid.org/0000-0002-2608-3815
Malcolm F White (iD) https://orcid.org/0000-0003-1543-9342

### Decision letter and Author response

Decision letter https://doi.org/10.7554/eLife.55852.sa1
Author response https://doi.org/10.7554/eLife.55852.sa2

## Additional files

### Supplementary files

• Transparent reporting form

### Data availability

All data generated or analysed during this study are included in the manuscript and supporting files. Source data files have been provided for Figures 2, 3, 4 and 6.

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
