## [Decision Letter]

Thank you for submitting your article "The dynamic interplay of host and viral enzymes in type III CRISPR-mediated cyclic nucleotide signalling" for consideration by *eLife*. Your article has been reviewed by three peer reviewers, and the evaluation has been overseen by a Reviewing Editor and Gisela Storz as the Senior Editor. The following individuals involved in review of your submission have agreed to reveal their identity: Philip Kranzusch (Reviewer #1); Martin Lawrence (Reviewer #2); Rotem Sorek (Reviewer #3).

The reviewers universally agree that this work is nearly ready for publication. No additional experiments are requested.

Summary:

It is becoming increasingly clear that cyclic nucleotides play a critical role in the host response to a viral infection and that viruses have evolved mechanisms to intercept these signaling molecules. In this work, the authors focused on the synthesis of cyclic-oligo-adenylate (cA_4_) produced by the type-III CRISPR system in *Sulfolobus solfataricus*, the immune proteins that are activated by cA_4_, and viral anti-CRISPR proteins that degrade cA_4_. Biochemical data is used to build a kinetic model, which helps explains how cyclases and ring nucleases govern cyclic nucleotide levels and determine infection outcomes in virus-host conflict.

Essential revisions:

1) Rates of cA_4_ synthesis by the *S. solfataricus* Csm complex have been previously measured (Rouillon et al., 2018; Rouillon et al., 2019). These rates should be incorporated into the mathematical model to understand how the relative rates of cA_4_ synthesis and cA_4_ degradation may influence effector function dynamics.

2) The current model does not take into account previously observations from the lab demonstrating that cA_4_ synthesis wanes after ~10 min if no new activating RNA is delivered to the system. Please discuss this in the context of your model.

---

## [Author Response]

Essential revisions:1) Rates of cA_4_ synthesis by the *S. solfataricus* Csm complex have been previously measured (Rouillon et al., 2018; Rouillon et al., 2019). These rates should be incorporated into the mathematical model to understand how the relative rates of cA_4_ synthesis and cA_4_ degradation may influence effector function dynamics.

We agree that this would strengthen the model. We have determined the rate of cA_4_ synthesis (Figure 2—figure supplement 2) from previously published data (Rouillon et al., 2018) and have incorporated this into the model. We have also adjusted the kinetic parameters of our model to the reaction temperature of 70 °C, which better represents infection kinetics at the growth temperature of *S. solfataricus*. We have also increased the cellular RNA concentration to 1000 µM, from 100 µM in the previous model, which we believe better represents cellular conditions and means that RNA is never fully depleted in the simulations. For comparison and transparency, we include a supplementary version of Figure 6 with simulations starting at fixed cA_4_ concentrations (as in the original submission) carried with the current model as per our initial submission. See Author response images 1 and 2. Ultimately, the changes we have made, including incorporation of the rate of cA_4_ synthesis do not radically alter any of our previous conclusions.

**Author response image 2. respfig2:** 

2) The current model does not take into account previously observations from the lab demonstrating that cA_4_ synthesis wanes after ~10 min if no new activating RNA is delivered to the system. Please discuss this in the context of your model.

This query relates to point 1, and we have addressed this point in the new model, which now incorporates the time dependent synthesis of cA_4_, dependent on the concentration of target RNA.